



# Leads and lags between Antarctic temperature and carbon dioxide during the last deglaciation

Léa Gest[1], Frédéric Parrenin[1], Jai Chowdhry Beeman[1], Dominique Raynaud[1], Tyler J. Fudge[2], Christo Buizert[3], Edward J. Brook[3].

[1]Univ. Grenoble Alpes, CNRS, IRD, IGE, F-38000 Grenoble, France
[2]Department of Earth and Space Sciences, University of Washington, Seattle, WA 98195, USA
[3]College of Earth, Ocean, and Atmospheric Sciences, Oregon State University, Corvallis, OR 97331, USA

*Correspondence to*: F. Parrenin (frederic.parrenin@univ-grenoble-alpes.fr)

**Abstract.** To understand causal relationships in past climate variations, it is essential to have accurate chronologies of
paleoclimate records. The last deglaciation, which occurred from 18,000 to 11,000 years ago, is especially interesting, since
it is the most recent large climatic variation of global extent. Ice cores in Antarctica provide important paleoclimate proxies,
such as regional temperature and global atmospheric $CO_2$. However, temperature is recorded in the ice while $CO_2$ is recorded
in the enclosed air bubbles. The ages of the former and of the latter are different since air is trapped at 50-120 m below the
surface. It is therefore necessary to correct for this air-ice shift to accurately infer the sequence of events.

Here we accurately determine the phasing between East Antarctic temperature and atmospheric $CO_2$ variations during the
last deglacial warming based on Antarctic ice core records. We build a stack of East Antarctic temperature variations by
averaging the records from 4 ice cores (EPICA Dome C, Dome Fuji, EPICA Dronning Maud Land and Talos Dome), all
accurately synchronized by volcanic event matching. We place this stack onto the WAIS Divide WD2014 age scale by
synchronizing EPICA Dome C and WAIS Divide using volcanic event matching, which allows comparison with the high
resolution $CO_2$ record from WAIS Divide. Since WAIS Divide is a high accumulation site, its air age scale, which has
previously been determined by firn modeling, is more robust. Finally, we assess the $CO_2$ / Antarctic temperature phasing by
determining four periods when their trends change abruptly.

We find that at the onset of the last deglaciation and at the onset of the Antarctic Cold Reversal (ACR) period $CO_2$ and
Antarctic temperature are synchronous within a range of 210 years. Then $CO_2$ slightly leads by $165 \pm 116$ years at the end of
the Antarctic Cold Reversal (ACR) period. Finally, Antarctic temperature significantly leads by $406 \pm 200$ years at the onset
of the Holocene period. Our results further support the hypothesis of no convective zone at EPICA Dome C during the last
deglaciation and the use of nitrogen-15 to infer the height of the diffusive zone. Future climate and carbon cycle modeling
works should take into account this robust phasing constraint.

## 1   Introduction

During the last million years or so, large glacial-interglacial transitions, or deglaciations, are observed in the paleorecord to
occur approximately every 100 ka (Jouzel et al., 2007; Lisiecki and Raymo, 2005; Williams et al., 1997). The last
deglaciation, often referred as glacial termination 1 (T1), offers a case study for a large global climatic change (very likely in
the 3-8°C interval on the global scale, IPCC, 2013) probably initiated by an orbitally driven insolation forcing (Berger, 1978;
Hays et al., 1976). The canonical interpretation of this apparent puzzle is that insolation acts as pacemaker of the climatic
cycles and the amplitude of glacial-interglacial transitions is mainly driven by two strong climatic feedbacks : atmospheric
$CO_2$ and continental ice surface-albedo changes. However, the exact role of atmospheric $CO_2$ forcing during the last
deglaciation, as well as the mechanisms that control the $CO_2$ rise, are still a matter of debate. Reconstructing the phase
relationship (leads and lags) between $CO_2$ and the different climate variables during the last termination has become of



importance, and has a long history in ice core research (Barnola et al., 1991; Caillon et al., 2003; Parrenin et al., 2013; Pedro
et al., 2012b; Raynaud and Siegenthaler, 1993).

T1 offers an interesting deglacial scenario, since warming and $CO_2$ rise were not continuous. Indeed, near the end of the glacial-interglacial transition, the Antarctic warming made a break during about 2000 years and even became reversed, preceding the Younger Dryas (YD) cooling in the Northern Hemisphere. This period of cooling in Antarctica is called the Antarctic Cold Reversal (ACR) and is almost synchronous with a warm period in the North, called the Bølling–Allerød
(B/A) period. Such changes in trend are useful tie points to observe the leads and lags between atmospheric $CO_2$ and Antarctic temperature.

Ice sheets are exceptional archives of past climates and atmosphere composition. The local temperature is recorded in the isotopic composition of snow/ice (Jouzel et al., 2007; NorthGRIP project members, 2004) thanks to the so-called isotopic paleothermometer (Lorius and Merlivat, 1977; Johnsen et al., 1989). The concentration of continental dust in ice sheets is a
proxy of continental aridity, atmospheric transport intensity and surface snow accumulation (Lambert et al., 2012). Finally, air bubbles enclosed in ice sheets are on a whole almost direct samples of past atmosphere. For atmospheric $CO_2$ concentrations, the atmospheric integrity is preserved in Antarctic ice but not in Greenland ice, because the latter has much higher concentrations of organic material and carbonate dust (Raynaud et al., 1993). The long ice core record of $CO_2$ covers the last 800 ka and has been essentially obtained from the Vostok and EPICA Dome C ice core (Lüthi et al., 2008). This
record is of global significance. However, the age of the air bubbles is younger than the age of the surrounding ice since air is locked-in at the base of firn, ~100 m below the surface (on the East Antarctic plateau), at the Lock-In Depth (LID, Parrenin et al., 2012a). The firn is composed in its top part of a convective zone (CZ) where the air is mixed, and in its bottom part of a diffusive zone (DZ) where molecular diffusion takes place (Sowers et al., 1992).

For years, it was believed that at the initiation of the termination (around 18 ka B1950, thousands years before 1950 A.D.),
just after the Last Glacial Maximum (LGM), Antarctic temperature started to warm $800 \pm 600$ yr before the $CO_2$ increase (Monnin et al., 2001), implying that $CO_2$ was not the cause of the initial deglacial warming in Antarctica. This study was based on the EPICA Dome C (EDC) ice core record (Jouzel et al., 2001) and was using a firn densification model to determine the air chronology. However, this firn densification model was later shown to be in error by several centuries for low accumulation sites such as EDC and for glacial conditions (Loulergue et al., 2007; Parrenin et al., 2012a).
Two more recent works (Pedro et al., 2012; Parrenin et al., 2013), used stacked temperature records and improved estimates of the relative age scale for the concentration of atmospheric $CO_2$ and Antarctic temperature. The first one (Pedro et al., 2012a) used the higher accumulation ice cores at Siple Dome and Law Dome, decreasing the uncertainty in the ice-air age shift. It concluded that $CO_2$ lagged Antarctic temperature by 0-400 yr in average during the last deglaciation. The second study (Parrenin et al., 2013) used the low accumulation EDC ice core but circumvented the use of firn densification models
by using nitrogen-15 as a proxy of the DZ height, and assuming no CZ. It concluded that $CO_2$ and Antarctic temperature were roughly in phase at the beginning of TI and at the end of the ACR period, but that $CO_2$ lagged Antarctic temperature by several centuries at the beginning of the Antarctic Cold Reversal and Holocene periods.

More recently (Marcott et al., 2014) published an unprecedented high-resolution $CO_2$ record from the West Antarctic Ice Sheet (WAIS) divide ice core, revealing centennial-scale changes in the global carbon cycle during the last deglaciation.
Moreover, the air chronology of WAIS Divide is well constrained thanks to a relatively high accumulation rate and to accurate nitrogen-15 measurements (Buizert et al., 2015). The deglacial temperature rise seen at WAIS Divide (WD) is similar to that at other Antarctic sites, with the exception that it shows early warming starting around 21 ka B1950, following local insolation (Cuffey et al., 2016; WAIS Divide Project Members and others, 2013). This early warming trend is not seen as strongly in records from East Antarctic ice cores such as EDC, Dome Fuji (DF), Vostok (VK), EPICA Dronning Maud
Land (EDML) and Talos Dome (TALDICE) (WAIS Divide Project Members and others, 2013). This could be representative



of the West Antarctic climate. However, the temperature rise at WAIS Divide shows an increased warming around 17.8 ka
B1950 like the other East Antarctic ice cores.

In the present work we revise our knowledge of leads and lags between Antarctic temperature and $CO_2$, using (i) a new stack
of accurately synchronized Antarctic temperature records, (ii) an accurate volcanic match between EDC and WAIS Divide
allowing the stack to be placed on the WD2014 chronology, (iii) the $CO_2$ record and air chronology from WD and (iv) an
improved method to determine the changes of trend.

## 2    Data and Methods

### 2.1    The Antarctic Temperature Stack (ATS2)

When investigating the phase relationship between the ice record of atmospheric $CO_2$ and Antarctic temperature it is
important to consider a stack of temperature records in order to remove local influences and noise in the individual records to
the greatest extent possible. The Antarctic Temperature Stack (ATS) provided by Parrenin et al., (2013) has been obtained by
averaging records from different locations in East Antarctica, which includes the EDC, DF, TALDICE, EDML and VK ice
cores. These records have been all synchronized to the EDC one by using isotopic or volcanic matching.

Here we improved the ATS of Parrenin et al. (2013) by: 1) removing the Vostok record, which was not accurately
synchronized with EDC because the volcanic and the isotopic matches have been done on two different cores taken on the
Vostok site : the VK-5G core (for isotopic record) and the VK-3G core (for volcanic record); 2) by using a volcanic
synchronization between DF and EDC (Fujita et al., 2015), previously synchronized with isotopes. 3) by including the WD
core (WAIS Divide Project Members and others, 2013), which brings West Antarctic climate variations into the ATS, which
was previously lacking.

The comparison between the two stacks is displayed on Figure 1, together with the difference between ATS and ATS2.
Qualitatively, ATS and ATS2 are in very good agreement, displaying the same temperature variations even at a centennial
time-scale. Quantitatively, there is less than 0.5°C variations between both stacks.

### 2.2    Placing EDC and ATS2 onto the WD2014 age scale

To analyse the $CO_2$ and ATS2 phase relationship, it is necessary to place them on the same chronology. Here we use the
WD2014 chronology as a reference (Buizert et al., 2015; Sigl et al., 2016). We therefore need to transfer ATS2 onto this
chronology. To achieve this, we derived a volcanic synchronization between EDC and WD ice cores (Figure 2). The method
of volcanic synchronization is explained in Parrenin et al. (2012b). Here, we used the Electrical Conductivity Measurements
(ECM). ECM data bring to light volcanic horizons by the detection of ice acidity. Acidity in the ice indicates volcanic
fallout, which is found in several paleorecords including ice cores. It enables us to detect volcanic peaks in different ice
cores, and by extension, the synchronisation between WAIS Divide and EDC ice core. Data from WAIS Divide were
collected (Tyler J. Fudge, 2014) and are divided in two different datasets : raw data from 6.4 to 11.4 ka B1950, and adjusted
data after 11.4 ka B1950. Up to 11.4 ka B1950 the data were well-resolved and there was no reason to make adjustments.
From 11.4 ka B1950, Fudge (2014) started making multiple tracks along the WD ice core which needed further treatments.
Normalization was performed by subtracting the mean conductivity of the section, then dividing by three times the standard
deviation. The mean and standard deviations were calculated from a subset of the data to prevent volcanic signals from
obscuring the annual signal after the data have been normalised (Tyler J. Fudge, 2014). EDC data were published by Udisti
et al. (2004).



We first determined a set of obvious major tie points and adjusted the time scale by taking them into account. Then, a set of
secondary minor tie points were determined in between. In the end, we obtained 144 tie points. We regularized the

synchronisation using the duration ratio between EDC and WD ice age between two consecutive tie points (orange curve on
the lower part of the Figure 2). Indeed, keeping this curve as smooth as possible amounts to minimizing the necessary
distortions in the EDC chronology. The 2σ uncertainty of this synchronisation (plotted in green on the second panel of Figure
2) is estimated as 20% of the distance to the nearest tie point (Parrenin et al., 2007b). The uncertainties stay roughly under 50
years until 13.5 ka B1950, and under 100 years after that. These uncertainties are taken into account in the uncertainties of

time delay determination. More detailed figures of the synchronization are available in supplementary materials.

### 2.3 The $CO_2$ record from WD onto the WD2014 air age scale

The atmospheric $CO_2$ data from the WD ice core have been published by Marcott et al. (2014). They consist in 1,030
measurements between 23,000 and 9,000 years B1950 with a median resolution of ~25 years. There was at least one
duplicate sample per depth and the mean standard deviation was ~1 ppm. There is an unexplained ~4 ppm systematic

positive offset of the WD $CO_2$ record compared to other ice core $CO_2$ records such as EDC (Parrenin et al., 2013), Siple
Dome and Law Dome (Pedro et al., 2012a).

$CO_2$ is measured in the air bubbles entrapped into the ice core while temperature is recorded through the isotopic content of
the ice. Because of the late closure of the air bubbles inside the firn, the air is younger than the surrounding ice at a given
depth. This offset is called $\Delta_{age}$. At WD, $\Delta_{age}$ is calculated using a firn densification model, which is constrained using

nitrogen-15 data, a proxy for firn column thickness (Buizert et al., 2015). $\Delta_{age}$ ranges from ~500±100 yr at the last glacial
maximum, to ~200±30 yr during the Holocene.

### 2.4 Break points determination by the LinearFit method

Parrenin et al. (2013) used a Metropolis Monte-Carlo search around a previous visually determined fit to find the best
agreement to the data. Tests were made of a modification of this method which eliminates the need for an initial fit.

However, this method was not suitable because the centenial atmospheric events identified by Marcott et al. (2014) dominate
the high resolution atmospheric $CO_2$ data, making it difficult to identify points in the LinearFit method that are common to
both $CO_2$ and temperature series.

We opt instead for a nonlinear least squares algorithm to fit our new data; keeping the initial visually determined fit (all
given in supplementary materials). This allows for the identification of a local optimal fit which is able to represent the

common change points between $CO_2$ and temperature. We improved upon this method in the following ways:

- We take both measurement and modelling uncertainties into account to determine the best fit. The measurement
error of each data point is given a priori. We estimate modelling error as the standard deviation of the residuals to
the initially determined fit.
- In the least-squares formulation of Parrenin et al. (2013), it is assumed that no correlation exists in the residuals.

This is not optimal, since it is clear from Figure 3 that consecutive residuals are correlated. To formally take this
correlation into account, we include the inverse of an estimate of the error autocorrelation matrix when calculating
the residuals (following Parrenin et al., 2015, for example). This estimate is calculated using the residuals of an
initial best fit, which we calculate assuming the errors to be uncorrelated.

In Figure 3, the optimal fit is shown, together with the uncertainties in the timings of the break points (the initial fit values

are given in the table S1 in Suppl. Mat.). We also computed the break points with the Mudelsee Breakfit algorithm
(Mudelsee, 2009) and obtained results in agreement within the uncertainty range (see Table 1). However, our method is
equally robust, mathematically simpler and less subjective. Indeed, the LinearFit method allows the dataset to be used as a



whole, where the BreakFit algorithm requires the user to delimit several time intervals introducing subjectivity in the determination of the break points. The bootstrap resampling used in the BreakFit algorithm to estimate modelling uncertainty
is robust and statistically correct but both more complicated and more parameter dependant than our method.

### 2.5    The CH₄ records from WD and EDC onto the WD2014 air agescale

The atmospheric $CH_4$ data measured on the WD ice core were measured using a laser spectrometer and a continuous flow analysis (CFA) method (Rhodes et al., 2015), which allows for a ~5 cm resolution (Rhodes et al., 2013). At EDC, the discrete measurements have been obtained by gas chromatography (Loulergue et al., 2008), with a resolution of several
meters.

As previously explained, $\Delta_{age}$ at WD is calculated using a firn densification model, which is constrained with nitrogen-15 data (Buizert et al., 2015). At EDC, $\Delta_{depth}$, the depth shift between synchronous air and ice levels, is calculated using 1) an estimate of the LID based on nitrogen-15 data (Dreyfus et al., 2010) and assuming a zero convective zone, 2) an average firn relative density (density divided by the density of pure ice) of 0.698 (Parrenin et al., 2012a) and 3) an estimate of the vertical
thinning function based on a 1D age/velocity model (Parrenin et al., 2007a).

### 3    Results and discussion

Our standard LinearFit run uses 6 points for $CO_2$ and Antarctic temperature, marking the deglacial onset, the onset and end of the ACR and the onset of the Holocene. We found that $CO_2$ and Antarctic temperature are synchronous within uncertainties at the beginning of the deglaciation around 17.8 ka B1950 ($34 \pm 210$ years lag of $CO_2$) and the onset of the
Antarctic Cold Reversal at 14.7 ka ($100 \pm 133$ years lag of $CO_2$). The LinearFit method computed two significant time delays, at the ACR end ($165 \pm 116$ years lead of $CO_2$) and the Holocene onset ($406 \pm 200$ years lag of $CO_2$). These time delays are identified on the Figure 3 through the vertical light blue ($CO_2$) and pink (ATS2) lines, with their corresponding values.

Our results generally confirm the results of Parrenin et al. (2013), with no significant lead/lag of $CO_2$ and Antarctic
temperature at the onset of the last deglacial warming and with a significant lag of $CO_2$ at the onset of the Holocene period. It is an important result, since the use of nitrogen-15 as a proxy of the firn height was subject to debates. Here, we use a different and more robust approach by using the ice-air shift at WD which is far better constrained than at EDC. However, our results show differences with Parrenin et al. (2013) at the onset of the  Antarctic Cold Reversal where Parrenin et al. found a $260 \pm 130$ year lag of $CO_2$ (taking into account the uncertainty in the determination of $\Delta_{depth}$), and at the ACR end
where no significant time delay between ATS and atmospheric $CO_2$ was found. The non-significant differences between ATS and ATS2 (Figure 1) indicate that our updated atmospheric $CO_2$ dataset is responsible for the different time delays computed. This testifies of the importance of the data resolution measurements in the determination of leads and lags. Until now, only links on millenial time scale between ATS2 and $CO_2$ were determined, but the growing resolution of these data also revealed a centennial time-scale relationship. Indeed, both of the 16.3 and 14.8 abrupt $CO_2$ rises (Marcott et al., 2014) are clearly
visible on the ATS2 signal (Figure 3).

We performed 2 other LinearFit tests of the data with different numbers of break points. To capture the entire deglacial warming, disregarding the middle transitions, we ran the program with 4 points. We also used 10 points to take into account the Marcott et al. (2014) centennial events, adding points around 16.1 and 14.7 for the mid-deglacial event, 15.9 for the pre-Bølling–Allerød event and 11.9 ka B1950 for the pre-Holocene event. The results are given in the Table 2 and Figure 4. Both
results from the 10 and 6 point LinearFit are consistent inside the uncertainty range. Given the variability of the data during the deglaciation, the 4 point LinearFit results are quite different, especially at the Holocene onset with a non significant $CO_2$ lead of $207 \pm 271$ yr.




Shakun et al. (2012) studied the last deglaciation and the time delays between an Antarctic temperature stack of Law Dome, Byrd, Siple Dome, EDML and Talos Dome records (Pedro et al., 2011), $CO_2$ concentrations from EDC and a global
temperature stack of 80 temperature records. They found that $CO_2$ generally leads global temperature, but slightly lags Antarctic temperature (Figure 5). From this exercise, Shakun et al. (2012) concluded that $CO_2$ was not the initial cause of the deglaciation, since it started increasing after Antarctic temperature, but still an important driver of the global temperature. But Shakun et al. (2012) used the EDC $CO_2$ record dated using firn densification models (Lemieux-Dudon et al., 2010). These firn models have been proven to be inadequate for glacial conditions in central East Antarctica (Loulergue et al., 2007;
Parrenin et al., 2012a). Here, we show that $CO_2$ could even lead Antarctic temperature, making it a potential cause of the deglacial warming in Antarctica.

The WD and EDC methane records onto the WD2014 age scale are plotted on Figure 6. Looking at the three fast methane transitions at the onset of the B/A, onset of YD and onset of the Holocene, we note that both age scales are compatible, although the EDC methane resolution sometimes does not allow for an accurate check (namely for the B/A onset transition).
This supports the approach taken by Parrenin et al. (2013) based on nitrogen-15 data to estimate the DZ and the assumption of a zero CZ during T1. It is consequently not needed to involve a varying convective zone to explain the nitrogen-15 record at EDC (Dreyfus et al., 2010).

## 4    Conclusion

Our study is a follow-up of the studies by Pedro et al. (2012a) and Parrenin et al. (2013) on the lead and lag between
atmospheric $CO_2$ and Antarctic temperature during the last deglacial warming. We improved upon this last study by: 1) using the high resolution $CO_2$ record from WD, this ice core being volcanically synchronized onto the EDC one; 2) using the ice-air shift computed on WD; 3) deriving a new East Antarctic Temperature Stack of 5 ice cores only volcanically synchronized; 4) taking into account the autocorrelation of residuals in our break points determination method (LinearFit). As in Parrenin et al. (2013), we found no $CO_2$ / temperature phasing at the onset of the deglaciation and a significant lag of
$CO_2$ at the end of the deglaciation. We find no noticeable lead or lag at the onset of the Antarctic Cold Reversal period while Parrenin et al. (2013) found a lag of $CO_2$, and a lead of $CO_2$ at the end of the ACR period where Parrenin et al. (2013) found no lead nor lag. We also confirm the ice-air depth shift computed at EDC by Parrenin et al. (2013), assuming the nitrogen-15 data represent the Lock-In Depth. In particular, we confirm that no convective zone existed at EDC during this time period. Our study provides new constraints for climate and carbon cycle models.

## 5    Acknowledgements

We thank Michael Sigl for his support and great help to discuss this work; Mirko Severi for his EPICA Dome C data and for his support with the volcanic synchronisation. This work is supported by the French Fondation Ars et Cuttoli.

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





|  | LinearFit | Uncertainty | BreakFit | Uncertainty |
|---|---|---|---|---|
| Holocene onset | −406 | 200 | −463 | |
| ACR end | 165 | 116 | 173 | 128 |
| ACR onset | −100 | 133 | −284 | 293 |
| Deglaciation onset | −34 | 210 | −419 | 313 |

**Table 1: Leads and lags comparison between the LinearFit method and the BreakFit method. The results are in agreement within the uncertainty range. A negative value means a lag of $CO_2$.**


|  | pts nb | Holocene onset | | ACR end | | ACR onset | | Deglacial onset | |
|---|---|---|---|---|---|---|---|---|---|
|  | | Age WD2014 | stddev | Age WD2014 | stddev | Age WD2014 | stddev | Age WD2014 | stddev |
| $CO_2$ | 4 | 11153 | 328 | - | - | - | - | 17791 | 250 |
| | 6 | 11211 | 193 | 12921 | 104 | 14338 | 119 | 17687 | 171 |
| | 10 | 11342 | 51 | 12955 | 129 | 14392 | 50 | 17665 | 170 |
| ATS2 | 4 | 11617 | 123 | - | - | - | - | 17847 | 190 |
| | 6 | 11617 | 51 | 12756 | 51 | 14438 | 59 | 17720 | 122 |
| | 10 | 11605 | 37 | 12737 | 56 | 14439 | 47 | 17681 | 110 |

**Table 2: Age transitions from the LinearFit method with 4, 6 and 10 points using WD $CO_2$ and ATS2 datasets. The 4 points LinearFit represent the entire deglacial warming from LGM to Holocene, 6 points also takes into account the ACR period, and the 10 points LinearFit uses the 3 centennial events from Marcott et al. (2014).**

20




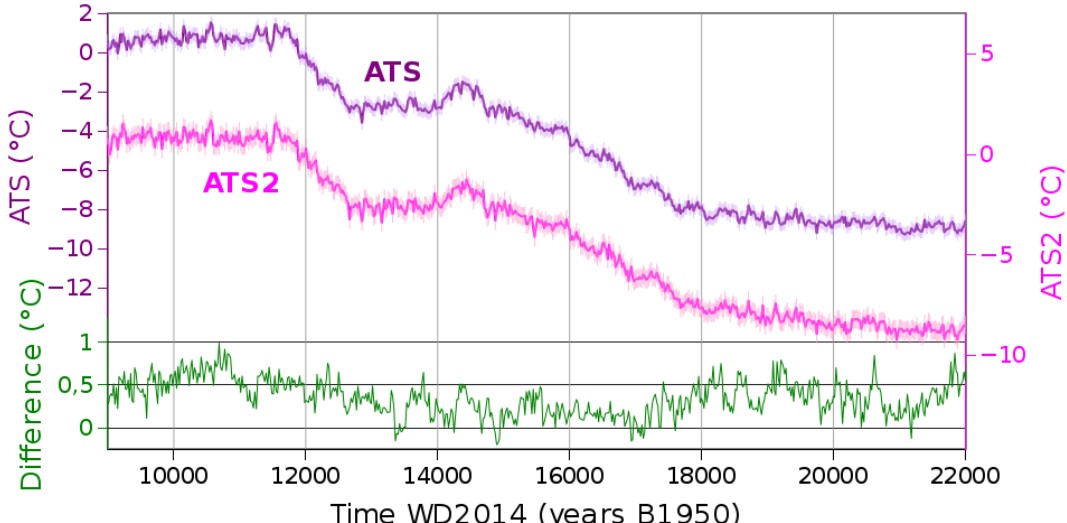

Figure 1: Antarctic Temperature Stack (ATS; purple), East Antarctic Temperature Stack (ATS2; pink), and the difference between ATS and ATS2 (green).



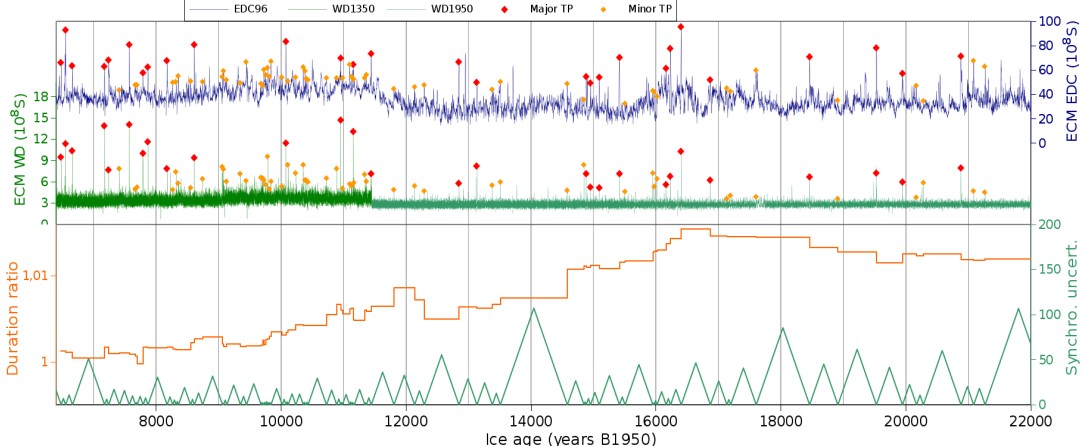

**Figure 2: Volcanic synchronisation between the EDC and WD ice cores. (Top) ECM records from EDC (blue) and WD (raw data: 6.4-11.4 ka; adjusted data: 11.4-24 ka). Red diamonds are major tie points, while orange diamonds are minor tie points. (Bottom) Ratio of the age difference between two consecutive tie points (orange) and uncertainty in the synchronisation (green) determined as 20% of the distance to the nearest tie point.**

235



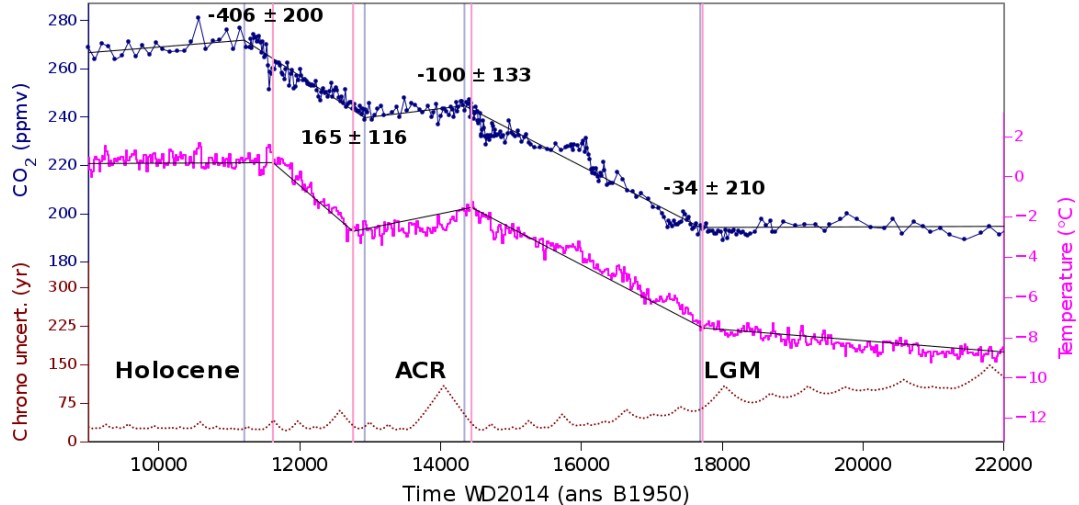

**Figure 3:** Atmospheric CO₂ (blue) and Antarctic temperature (pink) placed on a common time scale, with the relative uncertainty given below (red dotted line). The black lines represent the best LinearFit results of both records with 6 points. The blue and pink vertical lines mark the positions of the break points. A negative value means a lag of CO₂.



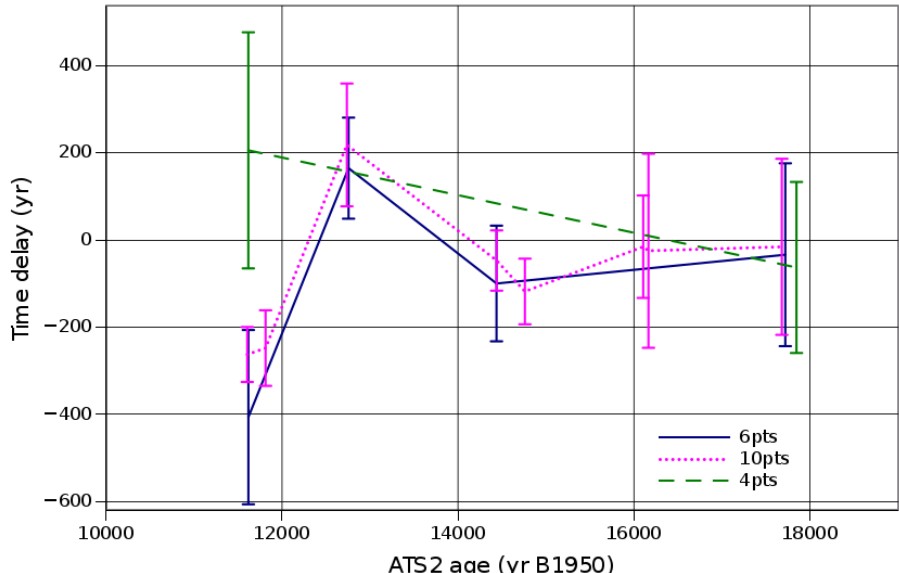

**Figure 4: 4 points LinearFit time delay (green), 6 points LinearFit time delay (blue) and 10 points LinearFit time delay (pink) between ATS2 and CO₂. A positive value means a lead by CO₂, a negative one means a lead by ATS2.**





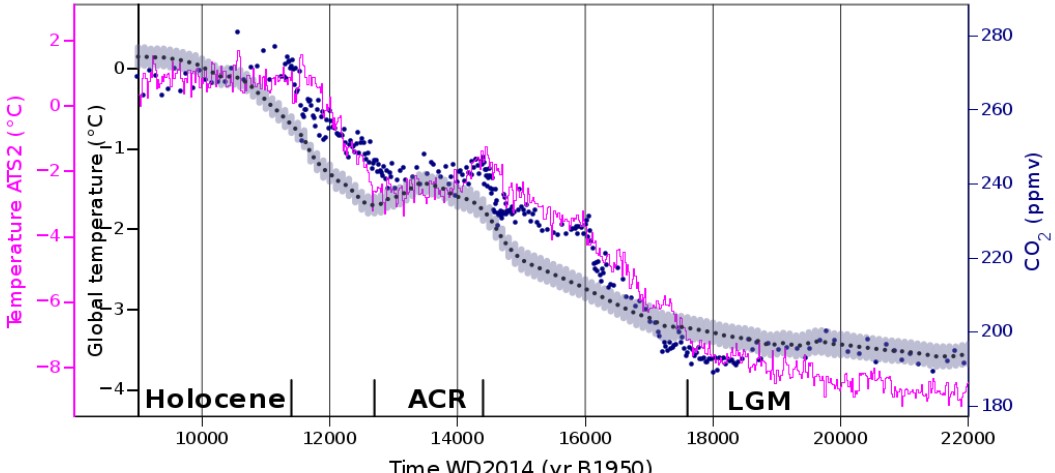

**Figure 5: Shakun et al. (2012) global temperature (black dots), ATS2 (pink curve) and WD $CO_2$ data (dark blue).**

30

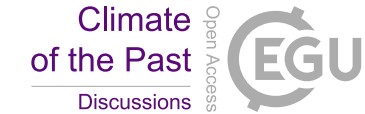



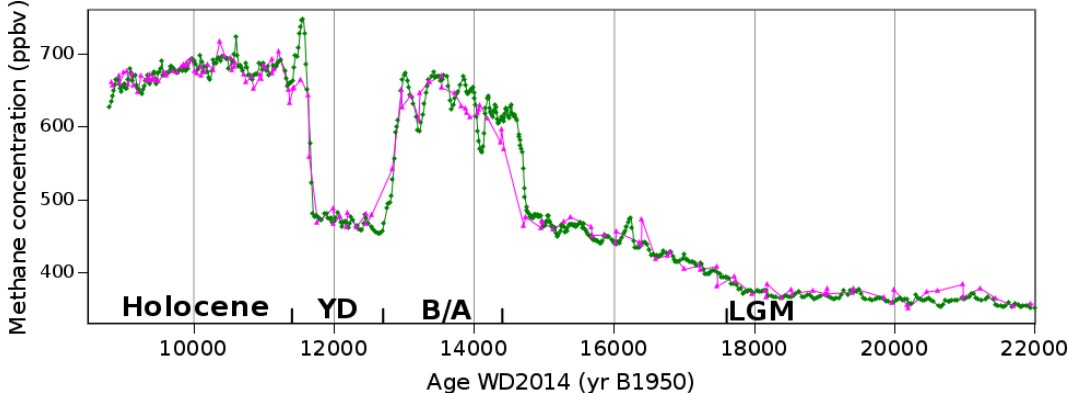

**Figure 6: Methane records at WD and EDC on the WD2014 agescale (the WD data were shifted downward by 10 ppbv for a better visual agreement). $\Delta_{age}$ at WD (Buizert et al., 2015) is an order of magnitude smaller than at EDC (Parrenin et al., 2013). The agreement between both chronologies at times of fast methane transitions confirms the validity of the approach taken by Parrenin et al. (2013) at EDC, in particular the assumption of a zero convective zone.**