# Peer review of "Leads and lags between Antarctic temperature and carbon dioxide during the last deglaciation"

_Climate of the Past, 2017_

## Referee Comment (RC1) · Anonymous Referee #1 · 25 Jul 2017

**General Comments**

The manuscript reports on the time relationship between atmospheric CO2 and Antarctic temperature during the last deglaciation. The authors make use of the WAIS CO2 record, which is the highest resolution and best-dated of the Antarctic CO2 records, and a new regional Antarctic temperature reconstruction from a stack of 4 Antarctic ice cores. Their central estimates of the temperature and CO2 time lag does not substantially differ from Pedro et al., 2012 and Parrenin et al 2013. However, their good methodology and use of the best records currently available leads to lower uncertainties and likely more robust results than those previous works.

The Antarctic temperature and CO2 time relationship is an important constraint for process studies of carbon cycle feedbacks during deglaciation so the new results are

valuable (even if there is little discussion of their significance in the manuscript itself). I believe the manuscript is well suited to publishing in Climate of the Past after revisions described below.

**Specific Comments**

The results should be presented more quantitatively in the abstract. For example the meaning of line 24 'synchronous within a range of 210 years' is unclear. The value at the onset of deglaciation is particularly important and must be reported in the abstract. Please report the central estimate of the lag for each breakpoint, the uncertainty, and whether these 1 sigma or 2 sigma uncertainties.

A mean lag over the deglaciation should also be given. This could be done using lag correlation analysis, as in Pedro et al., 2012. I'd suggest to exclude the onset of the Holocene from such an analysis since 1) the correlation between CO2 and the temperature stack appears to deteriorate there (can the authors confirm or reject this?) and 2) there is evidence that different carbon cycle mechanisms operate in the early Holocene compared to earlier in the deglaciation [e.g. Schmitt, Science, 2012]. If the authors will not provide an estimate of the mean lag over the deglaciation then they should provide a convincing argument as to why not.

It would strengthen the manuscript if the phasing analysis was also carried out on a radiative forcing time series determined from the WAIS CO2 record. Use of radiative forcing in place of the CO2 data may affect the results (e.g. see Ganopolski and Roche et al., 2009). It should be straightforward for the authors to do this since it was done in Parrenin et al., 2013.

Section 2.1: The ATS2 should either be made available as a Supplement, or archived at a public database.

In general the manuscript would benefit from a proof reading and edit from one of the coauthors who's first language is English.

**Technical Comments**

Line 12: ..phasing between CO2 and Antarctic temperature..

Line 28: "Future climate and carbon cycle modeling works should take into account this robust phasing constraint." Which of your constraints should the modelers use? A little more discussion would be appropriate.

Line 35: no space before colon.

Line 37: "..still a matter of debate". Agreed, but provide a citation.

Line 40: for completeness add Fischer et al., (1996) and Mudelsee, (2001).

Lines 41-46: This paragraph includes important information but is convoluted, please rewrite more clearly. Please cite the work supporting that the ACR and B-A are synchronous.

Line 55: you mean it represents global atmospheric CO2 variations?

Line 71: "roughly in phase".. be more precise.

Line 75: gas chronology

Line 98: Do you use the borehole calibrated WAIS temperature reconstruction of Cuffey et al., 2016? If so cite it, if not, why not?

Line 123: Explain why this is a reasonable error estimate.

Line 123: Roughly? Please be more precise.

Line 127: consist of

Line 129–131: Cite the original authors of the CO2 records. Note that Pedro et al., uses CO2 data from Byrd and Siple Dome, not Law Dome.

Line 136: 1 sigma or 2 sigma?

Line 141: Its unclear here if you are describing the Parrenin et al. (2013) method as LinearFit.

Line 143: "We opt instead for a nonlinear least squares.." this is confusing given you call your method 'LinearFit'.

Line 175: 1 sigma or 2 sigma uncertainties?

Line 187: to the importance

Line 219: "no CO2/temperature phasing"? Does not make sense, reformulate.

Lines 220–222: The way this is written is hard to follow. Suggest report your significant results and then report the important differences with respect to other studies.

**References**

Fischer, H., Wahlen, M., Smith, J., Mastroianni, D., and Deck, B.: Ice core records of atmospheric CO2 around the last three glacial terminations, Science, 283, 1712–1714, doi:10.1126/science.283.5408.1712, 1999.

Mudelsee, M.: The phase relations among atmospheric CO2 content, temperature and global ice volume over the past 420 ka, Quaternary Sci. Rev., 20, 583–589, doi:10.1016/S0277- 3791(00)00167-0, 2001.

Schmitt, J., Schneider, R., Elsig, J., Leuenberger, D., Lourantou, A., Chappellaz, J., Köhler, P., Joos, F., Stocker, T. F., Leuenberger, M., and Fischer, H.: Carbon isotope constraints on the deglacial CO2 rise from ice cores, Science, 336, 711–714, doi:10.1126/science.1217161, 2012.

---

## Referee Comment (RC2) · Anonymous Referee #2 · 18 Sep 2017

General remarks: This manuscript presents the phasing (lead/lag) between the isotopic records of several Antarctic ice cores (stacked into one record) and atmospheric $CO_2$ concentration from the WAIS Divide ice core (WDC). Essentially, this updated a previous result by the same group by making more robust age controls. It is important for documenting the phasing between Antarctic temperature proxy and atmospheric $CO_2$ concentration over glacial cycles for investigating the mechanisms of carbon cycle changes and their relation to the climate, and this manuscript could potentially contribute significantly as the most robust result.

However, I have a strong doubt about one of the resulting phasing, at the onset of the Holocene, as follows. I ask the authors to re-think the appropriateness of the employed method for obtaining meaningful phase information, if long-term trend is disturbed by

abrupt change around the end of the trend (as seen in $CO_2$ at the Holocene onset).

I could understand that the method employed here produces the lag of $CO_2$ by 400 yr "objectively", but visual inspection into Fig 2 actually shows different shapes of $CO_2$ and ATS2 signals, questioning the applicability of the simple breakpoint detection by line fitting in the first place. As discussed by Marcott et al. and repeated in this manuscript, abrupt (centennial-scale or less) changes in atmospheric $CO_2$ is important, and one of the major abrupt changes occurs at the onset of the Holocene (or the end of Termination). It seems inappropriate to detect the breakpoint here as the crossing point of the two lines fitted to the millennial-scale trends, ignoring the abrupt increase of $CO_2$ at around 11500 kyBP (very close to the breakpoint in ATS2). No change in trend is actually found at the 11211 yrBP (there is no significant change in linear trend from $\sim$9000 to $\sim$11500 yrBP). The trend line through the second increase of $CO_2$ over T1 goes near the lowest point in the earliest part around 13 ka and the highest point at around 11600 yrBP (just before abrupt rise), suggesting the overestimation of the slope detected for this long period due to the (automatic by method) inclusion of the abrupt $CO_2$ rise and subsequent high values. Thus, I suspect that the 406-yr lag of $CO_2$ is artifact by the method of fitting just two lines after 13 ka. The authors should consider if the method here for detecting the slope change is really appropriate, and if the statement in abstract that climate/carbon models should respect the phasing is reasonable, especially for the Holocene onset.

From this and specific comments below, I recommend the editor not to accept the manuscript in its current form. Thorough considerations on the method and results and another review round may be necessary.

Specific comments: L11. Proxy for temperature is recorded in ice (not the temperature itself).

L15. This time, it includes West Antarctic isotope record.

L16. stack of East.... <– Also West Antarctica.

L21. Add "for sites with much lower accumulation rates" after "firn modeling".

L25. See general remarks.

L27. Future climate.... I suggest deleting this sentence (see above).

L34. Add Abe-Ouchi et al., 2013.

L55. The description of the firn structure and the relation to the age of air are somewhat awkward. Please describe the three zones (convective, diffusive and lock-in zones) and lock-in depth in clear and compact manners. Add references for firn air studies at Antarctic inland sites; e.g. Bender et al., 1994 (GRL), Battle et al., 1996 (Nature), Kawamura et al., 2006 (EPSL), Landais et al., 2006 (QSR), Severinghaus et al., 2006 (EPSL).

L67. I think ice age-gas age difference should not be called in different ways than traditionally used (age "shift" does not sound right for me but please check with English speakers if you really want to use it).

L70. nitrogen-15 should be replaced by "isotopic ratio of N2 in air (d15N)" (and use d15N for the rest).

L100. ATS and ATS2 in Fig. 1 should both use WDC2014 age scale and it is indeed implied to be the case, but it does not seem to be explained in text. Also, it may be better to place the comparison of ATS and ATS2 after explaining the age scale.

L102. Quantitatively, .... I see that the amplitude of the difference between ATS and ATS2 is less than 0.5 degC, but the average is not zero. Please discuss the reason for the offset between the two stacks.

L111. This reference should be Fudge, 2014 (the position in the reference list should be wrong).

L112. "after". Actually, "before"?

L112. "Up to". Please clarify the range of age (e.g. "9 - 11.4 ka").

L.113. Here too, please clarify the range of age.

L.114. Why three times the S.D.? Please explain.

L.123. Why is the 20% reasonable? Please explain.

L.124. "after". <– "before"?

L.161. (section 2.5 as a whole) This comparison is not used for the ATS2-CO2 phasing estimation. It should be clearly spelled out and the aim of this comparison should be described in introduction.

L.176. See general remarks.

L.182-. The argument here (phasing was in error in 2013 paper because of low CO2 data resolution for EDC) requires the comparison between CO2 records from WD and EDC cores on the same time scale (here the CH4 comparison indeed becomes relevant to the central discussion of this study).

L.205. Here the authors should remind the readers that the Antarctic air temperature should not directly drive the atmospheric CO2, but it is the Southern Ocean which is thought to be mainly responsible for the CO2 glacial-interglacial variations, so it is important to further investigate in Antarctic ice cores for potential source temperature signals (i.e. Tsite from d-excess; Cuffey and Vimeux, 2001; Uemura et al., 2012).

Figure 1. Please show all individual Antarctic ice core records for ATS2 on WDC chronology. Drawing method in this figure and other figures are different (it is line between points in fig 1, and it is staircase function for other figures). Why?

Checking of English by native speakers (in authors) would be useful.

---

## Editor Comment (EC1) · H. Fischer (Editor) · 3 Oct 2017

Dear authors

your manuscript has now been seen by two expert reviewers, which both regard your paper as a significant contribution to the question of the phasing between gas (CO2) and ice (temperature) ice core records.

However, both reviewers also criticize your time series analytical approach for the end of the transition/beginning of the Holocene. Accordingly, in your reply to these reviewer comments, please include a detailed strategy how you can improve on or avoid this issue in a revised manuscript. This will be essential for the further process of the manuscript. Please also include a one-to-one reply to all other issues raised by the

reviewers.

Regards Hubertus Fischer (CP editor)